# A Meta-Learning Perspective on Cold-Start Recommendations for Items

**Manasi Vartak**[*]
Massachusetts Institute of Technology
mvartak@csail.mit.edu

**Arvind Thiagarajan**
Twitter Inc.
arvindt@twitter.com

**Conrado Miranda**
Twitter Inc.
cmiranda@twitter.com

**Jeshua Bratman**
Twitter Inc.
jbratman@twitter.com

**Hugo Larochelle**[†]
Google Brain
hugolarochelle@google.com

## Abstract

Matrix factorization (MF) is one of the most popular techniques for product recommendation, but is known to suffer from serious cold-start problems. Item cold-start problems are particularly acute in settings such as Tweet recommendation where new items arrive continuously. In this paper, we present a *meta-learning* strategy to address item cold-start when new items arrive continuously. We propose two deep neural network architectures that implement our meta-learning strategy. The first architecture learns a linear classifier whose weights are determined by the item history while the second architecture learns a neural network whose biases are instead adjusted. We evaluate our techniques on the real-world problem of Tweet recommendation. On production data at Twitter, we demonstrate that our proposed techniques significantly beat the MF baseline and also outperform production models for Tweet recommendation.

## 1 Introduction

The problem of recommending items to users — whether in the form of products, Tweets, or ads — is a ubiquitous one. Recommendation algorithms in each of these settings seek to identify patterns of individual user interest and use these patterns to recommend items. Matrix factorization (MF) techniques [19], have been shown to work extremely well in settings where many users rate the same items. MF works by learning separate vector embeddings (in the form of lookup tables) for each user and each item. However, these techniques are well known for facing serious challenges when making *cold-start* recommendations, i.e. when having to deal with a new user or item for which a vector embedding hasn't been learned. Cold-start problems related to *items* (as opposed to users) are particularly acute in settings where new items arrive continuously. A prime example of this scenario is Tweet recommendation in the Twitter *Home Timeline* [1]. Hundreds of millions of Tweets are sent on Twitter everyday. To ensure *freshness* of content, the Twitter timeline must continually rank the latest Tweets and recommend relevant Tweets to each user. In the absence of user-item rating data for the millions of new Tweets, traditional matrix factorization approaches that depend on ratings cannot be used. Similar challenges related to item cold-start arise when making recommendations for news [20], other types of social media, and streaming data applications.

In this paper, we consider the problem of item cold-start (ICS) recommendation, focusing specifically on providing recommendations when new items arrive continuously. Various techniques [3, 14, 27, 17]

---

[*]Work done as an intern at Twitter
[†]Work done while at Twitter

have been proposed in the literature to extend MF (traditional and probabilistic) to address cold-start problems. Majority of these extensions for item cold-start involve the incorporation of item-specific features based on item description, content, or intrinsic value. From these, a model is prescribed that can parametrically (as opposed to a lookup table) infer a vector embedding for that item. Such item embeddings can then be compared with the embeddings from the user lookup table to perform recommendation of new items to these users.

However, in a setting where new items arrive continuously, we posit that relying on user embeddings trained offline into a lookup table is sub-optimal. Indeed, this approach cannot capture substantial variations in user interests occurring on short timescales, a common phenomenon with continuously produced content. This problem is only partially addressed when user embeddings are retrained periodically or incrementally adjusted online.

Alternatively, recommendations could be made by performing *content-based filtering* [21], where we compare each new item to other items the user has rated in the recent past. However, a pure content-based filtering approach does not let us share and transfer information between users. Instead, we would like a method that performs akin to content filtering using a user's item history, but shares information across users through some form of transfer learning between recommendation tasks across users. In other words, we would like to learn a common procedure that takes as input a set of items from any user's history and produces a *scoring function* that can be applied to new test items and indicate how likely this user is to prefer that item.

In this formulation, we notice that the recommendation problem is equivalent to a *meta-learning problem* [28] where the objective is to learn a learning algorithm that can take as input a (small) set of labeled examples (a user's history) and will output a model (scoring function) that can be applied to new examples (new items). In meta-learning, training takes place in an episodic manner, where a training set is presented along with a test example that must be correctly labeled. In our setting, an *episode* is equivalent to presenting a set of historical items (and ratings) for a particular user along with test items that must be correctly rated for that user.

The meta-learning perspective is appealing for a few reasons. First, we are no longer tied to the MF model where a rating is usually the inner product of the user and item embeddings; instead, we can explore a variety of ways to combine user and item information. Second, it enables us to take advantage of deep neural networks to learn non-linear embeddings. And third, it specifies an effective way to perform transfer learning across users (by means of shared parameters), thus enabling us to cope with limited amount of data per user.

A key part of designing a meta-learning algorithm is the specification of how a model is produced for different tasks. In this work, we propose and evaluate two strategies for conditioning the model based on task. The first, called linear weight adaptation, is a light-weight method that builds a linear classifier and adapts weights of the classifier based on the task information. The second, called non-linear bias adaptation, builds a neural network classifier that uses task information to adapt the biases of the neural network while sharing weights across all tasks.

Thus our contributions are: (1) we introduce a new hybrid recommendation method for the item cold-start setting that is motivated by limitations in current MF extensions for ICS; (2) we introduce a meta-learning formulation of the recommendation problem and elaborate why a meta-learning perspective is justified in this setting; (3) we propose two key architectures for meta-learning in this recommendation context; and (4) we evaluate our techniques on a production Twitter dataset and demonstrate that they outperform an approach based on lookup table embeddings as well as state-of-the-art industrial models.

## 2   Problem Formulation

Similar to other large-scale recommender systems that must address the ICS problem [6], we view recommendation as a binary classification problem as opposed to a regression problem. Specifically, for an item $t_i$ and user $u_j$, the outcome $e_{ij} \in \{0, 1\}$ indicates whether the user *engaged* with the item. Engagement can correspond to different actions in different settings. For example, in video recommendation, engagement can be defined as a user viewing the video; in ad-click prediction, engagement can be defined as clicking on an ad; and on Twitter, engagement can be an action related to a Tweet such as liking, Retweeting or replying. Our goal in this context is to predict the probability

that $u_j$ will *engage* with $t_i$:

$$\Pr(e_{ij}{=}1|t_i, u_j) . \tag{1}$$

Once the engagement probability has been computed, it can be combined with other signals to produce a ranked list of recommended items.

As discussed in Section 1, in casting recommendations as a form of meta-learning, we view the problem of making predictions for one user as an individual task or episode. Let $T_j$ be the set of items to which a user $u_j$ has been exposed (e.g. Tweets recommended to $u_j$). We represent each user in terms of their **item history**, i.e., the set of items to which they have been exposed and their engagement for each of these items. Specifically, user $u_j$ is represented by their item history $V_j = \{(t_m, e_{mj})\} : t_m \in T_j$. Note that we limit item history to only those items that were seen before $t_i$.

We then model the probability of Eq. 1 as the output of a model $f(t_i; \theta)$ where the parameters $\theta$ are produced from the item history $V_j$ of user $u_j$:

$$\Pr(e_{ij}{=}1|t_i, u_j) = f(t_i; \mathcal{H}(V_j)) \tag{2}$$

Thus meta-learning consists of learning the function $\mathcal{H}(V_j)$ that takes history $V_j$ and produces parameters of the model $f(t_i; \theta)$.

In this paper, we propose two neural network architectures for learning $\mathcal{H}(V_j)$, depicted in Fig. 1. The first approach, called Linear Weight Adaptation (LWA) and shown in Fig. 1a, assumes $f(t_i; \theta)$ is a linear classifier on top of non-linear representations of items and uses the user history to adapt classifier weights. The second approach, called Non-Linear Bias Adaptation (NLBA) and shown in Fig. 1b, assumes $f(t_i; \theta)$ to be a neural network classifier on top of item representations and uses the user history to adapt the biases of the units in the classifier.

In the following subsections, we describe the two architectures and differences in how they model the classification of a new item $t_i$ from its representation $\mathcal{F}(t_i)$.

## 3   Proposed Architectures

As shown in Fig. 1, both architectures we propose take as input (a) the items to which a user has been exposed along with the rating (i.e. class) assigned to each item by this user (positive, i.e., 1, or negative, i.e., 0), and (b) a test item for which we seek to predict a rating. Each of our architectures in turn consists of two sub-networks. The first sub-network learns a common representation of items (historical and new), which we note $\mathcal{F}(t)$. In our implementation, item representations $\mathcal{F}(t)$ are learned by a deep feed-forward neural network. We then compute a single representative per class by applying an aggregating function $\mathcal{G}$ to the representations of items $t_m \in T_j$ from each class. A simple example of $\mathcal{G}$ is the unweighted mean, while more complicated functions may order items by recency and learn to weigh individual embeddings differently. Thus, the class representative embedding for class $c \in \{0, 1\}$ can be expressed as shown below.

$$R_j^c = \mathcal{G}(\{\mathcal{F}(t_m)\} : t_m \in T_j \wedge (e_{mj} = c)) \tag{3}$$

Once we have obtained class representatives, the second sub-network applies the LWA and NLBA approaches to adapt the learned model based on item histories.

### 3.1   Linear Classifier with Task-dependent weights

Our first approach to conditioning predictions on users' item histories has parallels to latent factor models and is appealing due to its simplicity: we learn a linear classifier (for new items) whose weights are determined by the user's history $V_j$.

Given the two class-representative embeddings $R_j^0, R_j^1$ described above, LWA provides the bias (first term) and weights (second term) of a linear logistic regression classifier as follows:

$$[b, (w_0 R_j^0 + w_1 R_j^1)] = \mathcal{H}(V_j) \tag{4}$$

where scalars $b, w_0, w_1$ are trainable parameters. More concretely, with $f(t_i; \theta)$ being logistic regression, Eq. 2 becomes:

$$\Pr(e_{ij}{=}1|t_i, u_j) = \sigma(b + \mathcal{F}(t_i) \cdot (w_0 R_j^0 + w_1 R_j^1)) \tag{5}$$

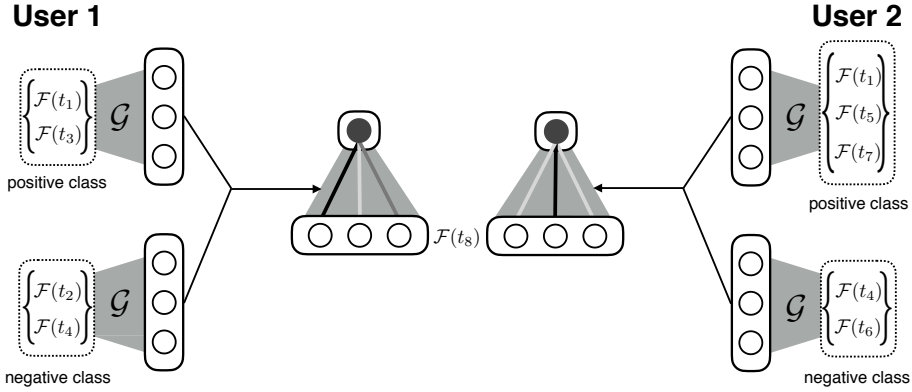

(a) Linear Classifier with Weight Adaptation. Changes in the shading of each connection with the output unit for two users illustrates that the weights of the classifier vary based on each user's item history. The output bias indicated by the shades of the circles however remains the same.

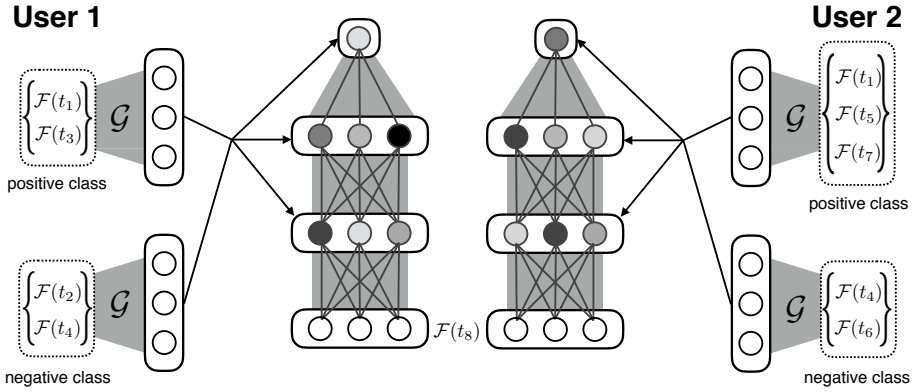

(b) Non-linear Classifier with Bias Adaptation. Changes in the shading of each unit between two users illustrates that the biases of these units vary based on each user's item history. The weights however remain the same.

Figure 1: Proposed meta-learning architectures

While bias $b$ of the classifier is constant across users, its weight vector varies with user-specific item histories (i.e., based on the representative embeddings for classes). This means that different dimensions of $\mathcal{F}(t_i)$, such as properties of item $t_i$, get weighted differently depending on the user.

While simple, the LWA method can be quite effective (see Section 5). Moreover, in some cases, it may be preferred over more complex methods because it allows significant amount of computation to be performed offline. For example, in Eq. 5, the only quantities that are unknown at prediction time are $\mathcal{F}(t_i)$. All the rest, including $R_j^c$, can be pre-computed, reducing the cost of prediction to the computation of one dot product and one sigmoid.

## 3.2 Non-linear Classifier with Task-dependent Bias

Our first meta-learning strategy is simple and is reminiscent of MF with non-linear embeddings. However, it limits the effect of task information, specifically $R_j^0$ and $R_j^1$, on the final prediction.

Our second strategy, NLBA, learns a neural network classifier with $H$ hidden-layers where the bias (first term) and weights (second term) of the *output*, as well as the biases (third term) and weights (fourth term) of *all hidden layers* are determined as follows:

$$[\mathbf{v}^0 R_j^0 + \mathbf{v}^1 R_j^1, \mathbf{w}, \{\mathbf{V}_l^0 R_j^0 + \mathbf{V}_l^1 R_j^1\}_{l=1}^H, \{\mathbf{W}_l\}_{l=1}^H] = \mathcal{H}(V_j) \qquad (6)$$

Here, the vectors $\mathbf{v}^0, \mathbf{v}^1, \mathbf{w}$ and matrices $\{\mathbf{V}_l^0\}_{l=1}^H, \{\mathbf{V}_l^1\}_{l=1}^H, \{\mathbf{W}_l\}_{l=1}^H$ are all trainable parameters. In contrast to LWA, all weights (output and hidden) in NLBA are constant across users, while the biases of output and hidden units are adapted per user. One can think of this approach as learning a shared pool of hidden units whose activation can be modulated depending on the user (e.g. a unit could be entirely shot down for a user with a large negative bias).

Compared to LWA, NLBA produces a non-linear classifier of the item representations $\mathcal{F}(t_i)$ and can model complex interactions between classes and also between the classes and the test item. For example, interactions allow NLBA to explore a different part of the classifier function space that is not accessible to LWA (e.g., ratio of the $k^{th}$ dimension of the class representatives). We find empirically that NLBA significantly improves model performance compared to LWA (Section 5).

**Selecting Historical Impressions.** A natural question with our meta-learning formulation is the minimum item history size required to make accurate predictions. In general, item history size depends on the problem and variability within items, and must be empirically determined. Often, due to the long tail of users, item histories can be very large (e.g., consider a bot which likes every item). Therefore, we recommend setting an upper limit on item history size. Further, for any user, the number of items with positive engagement ($e_{ij}$=1) can be quite different from those with negative engagement ($e_{ij}$=0). Therefore, in our experiments, we choose to independently sample histories for the two classes up to a maximum size $k$ for each class. Note that while this sampling strategy makes the problem more tractable, this sampling throws off certain features (e.g. user click through rate) that would benefit from having the true proportions of positive and negative engagements.

## 4 Related Work

Algorithms for recommendation broadly fall into two categories: *content-filtering* and *collaborative-filtering*. Content filtering [21] uses information about items (e.g. product categories, item content, reviews, price) and users (e.g. age, gender, location) to match users to items. In contrast, collaborative filtering [19, 23] uses past user-item ratings to predict future ratings. The most popular technique for performing collaborative filtering is via latent factor models where items and users are represented in a common latent space and ratings are computed as the inner product between the user and item representations. Matrix factorization (MF) is the most popular instantiation of latent factor models and has been used for large scale recommendations of products [19], movies [15]) and news [7]. A significant problem with traditional MF approaches is that they suffer from cold-start problems, i.e., they cannot be applied to new items or users. In order to address the cold-start problem, work such as [3, 14, 27, 17] has extended the MF model so that user- and item-specific terms can be included in their respective representations. These methods are called *hybrid methods*. Given the power of deep neural networks to learn representations of images and text, many of the new hybrid methods such as [30] and [12] use deep neural networks to learn item representations. Deep learning models based on ID embeddings (as opposed to content embeddings) have also been used for performing large scale video recommendation in [6].

The work in [5, 30] operates in a problem setting similar to ours where new scientific articles must be recommended to users based on other articles in their library. In these settings, users are represented in terms of scientific papers in their "libraries". Note that unlike our setting where we have positive as well as negative information, there are no *negative* examples present in this setting. [9, 10] propose RNN architecture for a similar problem, namely that of recommending items during short-lived web sessions. [11] propose co-factorization machines to jointly model topics in Tweets while making recommendations.

In this paper, we propose to view recommendations from a meta-learning perspective [28, 18]. Recently, meta-learning has been explored as a popular strategy for learning from a small number of items (also called *few-shot learning* [16, 13]). Successful applications of meta-learning include MatchingNets [29] in which an episodic scheme is used to train a meta-learner to classify images given very few examples belonging to each classes. In particular, MatchingNets use LSTMs to learn attention weights over all points in the support set and use a weighted sum to make predictions for the test item. Similarly, in [24], the authors propose an LSTM-based meta-learner to learn another learner that performs few-shot classification. [25] proposes a memory-augmented neural network for meta-learning. The key idea is that the network can slowly learn useful representations of data through weight updates while the external memory can cache new data for rapid learning. Most

recently, [4] proposes to learn active learning algorithms via a technique based on MatchingNets. While the above state-of-the-art meta-learning techniques are powerful and potentially useful for recommendations, they do not scale to large datasets with hundreds of millions of examples.

Our approach of computing a mean representative per class is similar to [26] and [22] in terms of learning class representative embeddings. Our work also has parallels to the recent work on DeepSets [31] where the authors propose a general strategy for performing machine learning tasks on *sets*. The authors propose to learn an embedding per item and then use a *permutation invariant* operation, usually a sumpool or maxpool, to learn a single representation that is then passed to another neural network for performing the final classification or regression. Our techniques differ in that our input sets are not *homogeneous* as assumed in DeepSets and therefore we need to learn multiple representatives, and unlike DeepSets, our network must work for variable size histories and therefore a weighted sum is more effective than the unweighted sum.

## 5 Evaluation

### 5.1 Recommending Tweets on Twitter Home Timeline

We evaluated our proposed techniques on the real-world problem of Tweet recommendation. The Twitter *Home timeline* is the primary means by which users on Twitter consume Tweets from their networks [1]. 300+ million monthly active users on Twitter send hundreds of millions of Tweets per day. Every time a user visits Twitter, the *timeline ranking algorithm* scores Tweets from the accounts they follow and identifies the most relevant Tweets for that user. We model the timeline ranking problem as one of engagement prediction as described in Section 2. Given a Tweet $t_i$ and a user $u_j$, the task is to predict the probability of $u_j$ engaging with $t_i$, i.e., $\Pr(e_{ij}=1|t_i, u_j; \theta)$. Engagement can be any action related to the Tweet such as Retweeting, liking or replying. For the purpose of this paper, we will limit our analysis to prediction of one kind of engagement, namely that of *liking* a Tweet. Because hundreds of millions of new Tweets arrive every day, as discussed in Section 1, traditional matrix factorization schemes suffer from acute cold-start problems and cannot be used for Tweet recommendation. In this work, we cast the problem in terms of meta-learning and adopt the formulation developed in Eq. 2.

**Dataset.** We used production data regarding Tweet engagement to perform an offline evaluation of our techniques. Specifically, the training data was generated as follows: for a particular day $d$, we collect data for all Tweet impressions (i.e., Tweets shown to a user) generated during that day. Each data point consists of a Tweet $t_i$, the user $u_j$ to whom it was shown, and the engagement outcome $e_{ij}$. We then join impression data with item histories (per user) that are computed using impressions from the week prior to $d$. As discussed in Section 2, there are different strategies for selecting items to build the item history. For this problem, we independently sample impressions with positive engagement and negative engagement, up to a maximum of $k$ engagements in each class. We experimented with different values of $k$ and chose the smallest one that did not produce a significant drop in performance. After applying other typical filtering operations, our training dataset consists of hundreds of millions of examples (i.e., $t_i$, $u_j$ pairs) for day $d$. The test and validation sets were similarly constructed, but for different days. For feature preprocessing, we scale and discretize continuous features and one-hot-encode categorical features.

**Baseline Models.** We implemented different architectural variations of the two meta-learning approaches proposed in Section 2. Along with comparisons within architectures, we compare our models against three external models: (a) first, an industrial baseline (PROD-BASE) not using meta-learning; (b) the industrial baseline augmented with a latent factor model for users (MF); and (c) the state-of-the-art production model for engagement prediction (PROD-BEST).

PROD-BASE is a deep feed-forward neural network that uses information pertaining only to the *current* Tweet in order to predict engagement. This information includes features of the current Tweet $t_i$ (e.g. its recency, whether it contains a photo, number of likes), features about the user $u_j$ (e.g. how heavily the user uses Twitter, their network), and the Tweet's author (e.g. strength of connection between the user and author, past interactions). Note that this network uses no historical information about the user. This baseline learns a combined item-user embedding (due to user features present in the input) and performs classification based on this embedding. While relatively simple, this model presents a very strong baseline due to the presence of high-quality features.

| Model | AUROC | AUROC (w/CTR) |
|---|---|---|
| MF (shallow) | +0.22% | +1.32% |
| MF (deep) | +0.55% | +1.87% |
| PROD-BEST | +2.54% | **+2.54**% |
| LWA | +1.76% | +2.43% |
| LWA* | +1.98% | +2.31% |

Table 1: Performance with LWA

| Model | AUROC | AUROC (w/CTR) |
|---|---|---|
| MF (shallow) | +0.22% | +1.32% |
| MF (shallow) | +0.55% | +1.87% |
| PROD-BEST | +2.54% | +2.54% |
| NLBA | +2.65% | **+2.76%** |

Table 2: Performance with NBLA

To mimic latent factor models in cold-start settings, in the second baseline, MF we augmented PROD-BASE to learn a latent-space representations of users based on ratings. MF uses an independently learned user representation and a current Tweet representation whose inner product is used to make the classification. We evaluate two versions of MF, one that uses a shallow network (1 layer) for learning the representations and another than uses a deep network (5 layers) to learn representations. PROD-BEST is the production model for engagement prediction based on deep neural networks. PROD-BEST uses features not only for the current Tweet but historical features as described in [8]. PROD-BEST is a highly tuned model and represents the state-of-art in Tweet engagement prediction.

**Experimental Setup.** All models were implemented in the Twitter Deep Learning platform [2]. Models were trained to minimize cross-entropy loss and SGD was used for optimization. We use AUROC as the evaluation metric in our experiments. All performance numbers denote **percent AUROC improvement** relative to the production baseline model, PROD-BASE. For every model, we performed hyperparameter tuning on a validation set using random search and report results for the best performing model.

## 5.2 Results

**Linear Classifier with Weight Adaptation.** We evaluated two key instantiations of the LWA approach. First, we test the basic architecture described in Section 3.1 where we calculate one representative embedding from the positive and negative class, and take a linear combination of the dot products of the new item with the respective embeddings. We note this architecture LWA (refer Fig. 1.a). When learning class representatives, we use a deep feed-forward neural network ($\mathcal{F}$ in Fig. 1.a) to first compute embeddings and then use a weighted average to learn class representatives ($\mathcal{G}$ in Fig. 1.a). We also evaluate a model variation where we augment LWA with a network that uses only the new item embedding to produce a prediction that is then combined with the prediction of LWA to produce the final probability. The intuition behind this model, called LWA*, is that the linear weight adaptation approach works well when there are non-zero items in the two engagement classes. In cases where one of the classes is empty, the model can fall back on the current item to predict engagement. We show the performance of all LWA variants in Table 1.

Note that for all models, we also test a variant where we explicitly pass a user-specific click-through-rate (CTR) to the model. The reason is that the CTR provides a strong prior on the probability $p(e_{ij} = 1)$ but cannot be easily learned from our architectures because the ratio of positive to negative engagements in the item history is not proportional to the CTR. User-specific CTR can be thought of as being part of the bias term from Eq. 2.

We find that the simplest classifier adaptation model, LWA, already improves upon the production baseline (PROD-BASE) by >1.5 percent. Adding the bias in the form of CTR, improves the AUROC even further. Because learning a good class representative embedding is key for our meta-learning approaches, we performed experiments varying only the architectures used to learn class representative embeddings (i.e., architectures for $\mathcal{F}, \mathcal{G}$ in Fig. 1a). The top-level classifier was kept constant at LWA but we varied the aggregation function and depth of the feed forward network used to learn $\mathcal{F}$. Results of this experimentation are shown in Table 3. We find that deep networks work better than shallow networks but a model with 9 hidden layers performs worse than a 5 layer network, possibly due to over-fitting. We also find that weighted combinations of embeddings (when items are sorted by time) perform significantly better than simple averages. A likely reason for the effectiveness of weighted combinations is that item histories can be variable sized; therefore, weighing non-zero entries higher produces better representatives.

**Non-Linear Classifier with Bias Adaptation.** As with LWA, we evaluated different variations of the non-linear bias adaptation approach. Results of this evaluation are shown in Table 2. We use a

| Hidden Layers | AVG | Weighted AVG |
| --- | --- | --- |
| 1 | +1.8% | +2.31% |
| 5 | +2.20% | +2.42% |
| 9 | +2.09% | +1.65% |

Table 3: Learning a representative per class

| Engagements Used | AUROC |
| --- | --- |
| POS/NEG | **+2.54%** |
| POS-ONLY | +1.76% |
| NEG-ONLY | +1.87% |

Table 4: Effect of different engagements

weighted mean for computing class representatives in NLBA. We see that this network immediately beats PROD-BASE by a large margin. Moreover, it also readily beats the state-of-the-art model, PROD-BEST. Augmenting NLBA with user-specific CTR further allows the network to cleanly beat the highly optimized PROD-BEST.

For NLBA architectures, we also evaluated the impact on model AUROC when examples of only one class ($e_{ij}$=0 or 1) are present in the item history. These architectures replicate the strategy of only using one class of items to make predictions. These numbers approximate the gain that could be achieved by using a DeepSets [31]-like approach. The results of this evaluation are shown in Table 4. As expected, we find that dropping either type of engagement reduces performance significantly.

**Summary of Results.** We find that both our proposed approaches improve on the baseline production model (PROD-BASE) by up to 2.76% and NLBA readily beats the state-of-the-art production model (PROD-BEST). As discussed in Sec. 3.2, we find that NLBA clearly outperforms LWA because of the non-linear classifier and access to a larger space of functions. A breakdown of NLBA performance by overall user engagement identifies that NLBA shows large gains for highly engaged users. For both techniques, although the improvements in AUROC may appear small numerically, they have large product impact because they translate to significantly higher number of engagements. This gain is particularly noteworthy when compared to the highly optimized and tuned PROD-BEST model.

## 6 Discussion

In this paper, we proposed to view recommendations from a meta-learning perspective and proposed two architectures for building meta-learning models for recommendation. While our techniques show clear wins over state-of-the-art models, there are several avenues for improving the model and operationalizing it. First, our model does not explicitly model the time-varying aspect of engagements. While weighting impressions differently is a way of modeling time dependencies (e.g., more recent impressions get more weight) scalable versions of sequence models such as [29, 10] could be used to explicitly model time dynamics. Second, while we chose a balanced sampling strategy for producing item histories, we believe that different strategies may be more appropriate in different recommendation settings, and thus merit further exploration. Third, producing and storing item histories for every user can be expensive with respect to computational as well as storage cost. Therefore, we can explore the computation of representative embeddings in an *online fashion* such that at any given time, the system tracks the $k$ most representative embeddings. Finally, we believe that there is room for future work exploring effective visualizations of learned embeddings when input items are not easy to interpret (i.e., beyond images and text).

## 7 Conclusions

In this paper, we study the recommendation problem when new items arrive continuously. We propose to view item cold-start through the lens of meta-learning where making recommendations for one user is considered to be *one task* and our goal is to learn across many such tasks. We formally define the meta-learning problem and propose two distinct approaches to condition the recommendation model on the task. The linear weight adaptation approach adapts weights of a linear classifier depending on task information while the non-linear bias adaptation approach learns task specific item representations and adapts biases of a neural network based on task information. We perform an empirical evaluation of our techniques on the Tweet recommendation problem. On production Twitter data, we show that our meta-learning approaches comfortably beat the state-of-the-art production models for Tweet recommendation. We show that the non-linear bias adaptation approach outperforms the linear weight adaptation approach. We thus demonstrate that recommendation through meta-learning is effective for item cold-start recommendations and may be extended to other recommendation settings as well.

## Addressing Reviewer Comments

We thank the anonymous reviewers for their feedback on the paper. We have incorporated responses to reviewer comments in the paper text.

## Acknowledgement

We would like to thank all members of the Twitter Timelines Quality team as well as the Twitter Cortex team for their help and guidance with this work.

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
