[Reviews · NeurIPS 2017]

Reviewer 1



This is an interesting and well-written paper but there are some parts that are not well explained, hence my recommendation. These aspects are not clear: 1. I am not sure about the "meta-learning" part. The recommendation task is simply formulated as a binary classification task (without using matrix factorization). The relation to meta-learning is not convincing to me. 2. "it becomes natural to take advantage of deep neural networks (the common approach in meta-learning)" - this is not a valid claim - deep learning is not the common approach for meta-learning; please see the papers by Brazdil and also the survey by Vilaltra & Drissi. 3. What is the input to the proposed 2 neural network architectures and what is its dimensionality? This should be clearly described. 4. I don't understand why in the first (liner) model the bias is constant and the weights are adapted and the opposite applies for the second (non-linear) model - the weights are fixed and the biases are adapted. This is an optimisation task and all parameters are important. Have you tried adaption all parameters? 5. The results are not very convincing - a small improvement, that may not be statistically significant. 6. Evaluation -Have you sampled the same number of positive and negative examples for each user when creating the training and testing data? -How was the evaluation done for items that were neither liked or disliked by a user? -Why are the 3 baselines called "industrial"? Are they the typically used baselines? -Is your method able to generate recommendations to all users or only for some? Is it able to recommend all new items? In other words, what is the coverage of your method? -It will be useful to compare you method with a a pure content-based recommender. Is any of the beselines purely content-based? 7. Discuss the "black-box" aspect of using neural networks for making recommendations (lack of interpretability) These issues need to be addressed and explained during the rebuttal.

Reviewer 2



This paper aims to address the item cold-start problem in recommender systems. The cold-start problem is a long-standing research topic in recommender systems. There is no innovation in the problem itself but the idea of using meta-learning to address this problem is novel to some degree. Two models, one shallow and one deep, are proposed for the task. Regarding the design of the method, I have a concern about the user part. Although the authors argue that one advantage of using meta-learning here is that we can explore a variety of ways to combine user and item information, they seem to lose some user information by an aggregating function. There are some missing related works for the item cold-start problem and tweet recommendation. Other than content filtering, there is another line of research for the cold-start problem which try to use active learning ideas to address this problem. The authors should be aware of them and provide a discussion in the related work section. Regarding tweet recommendation, there are also some previous works which treat it as a binary prediction problem or ranking problem and exploit different features (e.g. content). I think the factorization machine variants proposed in [Hong et al. Co-factorization machines: modeling user interests and predicting individual decisions in twitter. In WSDM, 2013] should be employed as a baseline for this work (for fairness, you can implement them based on the same features, i.e. user, item and item contents). It would be interesting to see if the method proposed here can beat them, considering they do not lose user information and some of them use ranking losses rather than point-wise losses. There are some typos, e.g. line 97 and line 101.

Reviewer 3



This paper proposes a cold-start recommendation system to recommend tweets to users. The approach is based on multi-task learning, where each user is a different task, and layers of a neural network are shared across users. Two alternatives are considered: one in which the bias terms are fixed across users, but the other weights can change; and one in which the bias term can change, and the other weights are fixed. The paper is fairly well-written and the approach appears sound. While there are some similar approaches (as described in the related work), the proposed variant appears novel, and large-scale experiments comparing against production algorithms suggest that the method improves accuracy. It is understandably difficult to give sufficient details of the existing production system, but without these details, it is hard to understand fully why the proposed approach works better. Please consider a more thorough error analysis to understand when and why the approach outperforms the baselines. Is it indeed performing better under "colder" conditions? Can any qualitative insight be provided by analyzing the learned user embeddings? What are the various hyperparameters tuned in Sec 4.1 and how sensitive is the method to this assignment? In summary, this appears to be a reasonable, if somewhat incremental, contribution to the recommendation community. To be a stronger contribution, it should provide insight as to when and why the approach works better than baselines.